# Rational Design of Alginate Lyase from *Microbulbifer* sp. Q7 to Improve Thermal Stability

**DOI:** 10.3390/md17060378

**Published:** 2019-06-25

**Authors:** Min Yang, Su-Xiao Yang, Zhe-Min Liu, Nan-Nan Li, Li Li, Hai-Jin Mou

**Affiliations:** College of Food Science and Engineering, Ocean University of China, Qingdao 266003, China; minyang89@163.com (M.Y.); yangsuxiao66@163.com (S.-X.Y.); ocean2013@126.com (Z.-M.L.); lxgcznlnn@163.com (N.-N.L.); llcs0229@163.com (L.L.)

**Keywords:** alginate lyase, thermal stability, rational design, disulfide bond, molecular dynamic simulation

## Abstract

Alginate lyase degrades alginate by the β-elimination mechanism to produce oligosaccharides with special bioactivities. The low thermal stability of alginate lyase limits its industrial application. In this study, introducing the disulfide bonds while using the rational design methodology enhanced the thermal stability of alginate lyase cAlyM from *Microbulbifer* sp. Q7. Enzyme catalytic sites, secondary structure, spatial configuration, and molecular dynamic simulation were comprehensively analyzed. When compared with cAlyM, the mutants D102C-A300C and G103C-T113C showed an increase by 2.25 and 1.16 h, respectively, in half-life time at 45 °C, in addition to increases by 1.7 °C and 0.4 °C in the melting temperature, respectively. The enzyme-specific activity and *k_cat_*/*K_m_* values of D102C-A300C were 1.8- and 1.5-times higher than those of cAlyM, respectively. The rational design strategy that was used in this study provides a valuable method for improving the thermal stability of the alginate lyase.

## 1. Introduction

Alginate lyases are derived from abundant sources, including bacteria, brown seaweed, and marine mollusks [1]. Alginate lyases are divided into seven polysaccharide lyase (PL) families (PL 5, 6, 7, 14, 15, 17, and 18) in the CAZy database (http://www.cazy.org/Polysaccharide-Lyases.html) based on the identification of amino acid sequences. Alginate lyase degrades alginate by the β-elimination mechanism to form a double bond between C4 and C5 at the non-reducing end [2]. The degradation products, alginate oligosaccharides, and 4-deoxy-L-erythro-5-hexoseulose uronic acid (DEH) are used for various functions in medical and energy industries [3,4,5]. Previous studies have demonstrated that alginate lyase exhibits poor thermal stability when being incubated with enzymes at above 40 °C for more than 0.5 h (residual enzyme activity < 50%) [6,7,8]. It is important for alginate lyase to maintain enzyme activity during the process of degrading alginate.

Protein engineering, including directed evolution and rational design, is an effective technology for improving enzyme properties [9]. Directed evolution is the construction of a mutant library by random mutagenesis, and the mutants are then screened according to the target performance. The enzyme activity of β-(1,4)-mannanases [10] and catalytic activity and thermal stability of tyrosine phenol-lyase [11] have been improved by directed evolution. The success of this method is highly dependent on the size and quality of the mutant library [12]. In contrast, rational design is an effective method for improving the enzyme properties by computer simulation-predicted mutation sites. The catalytic activity of XynB [12] and alginate lyase AlgL [13] and the thermal stability of alkaline pectate lyase [14], phytase [15], and alginate lyase NitAly [16] have been enhanced by rational design.

The disulfide bonds can increase protein stability by reducing the entropy of the protein unfolding process [17]. The introduction of disulfide bonds in a protein is a type of rational design that can effectively improve protein thermal stability [18]. The current strategy of introducing disulfide bonds in proteins, e.g., lipase B from *Candida antarctica* [19], PhyA from *Acidobacteria* [20], and alkaline α-amylase from *Alkalimonas amylolytica* [21], uses computational tools to predict the disulfide bonds, and then screens the mutants by molecular dynamic (MD) simulation. However, the computational tools often predict a high number of potential disulfide bonds, while failing to effectively identify the valid candidates and inactivated enzymes. Therefore, further information is required to precisely select the potential disulfide bonds to improve the enzyme thermal stability.

The alginate lyase cAlyM belonging to PL7 from *Microbulbifer* sp. Q7 has been expressed in *Escherichia coli*, and its particular properties have also been depicted [22]. However, cAlyM cannot meet the demand for industrial preparation of alginate oligosaccharides, owing to its low thermal stability. In this study, analyzing the catalytic motif, secondary structure characteristics, spatial configuration, and MD simulation screened the disulfide bonds. The mutants with disulfide bonds were obtained by site-directed mutagenesis. The properties and structure–function relationship of mutants with disulfide bonds were investigated, and the mutant D102C-A300C exhibiting significantly improved thermal stability was obtained.

## 2. Results

### 2.1. Selection of Potential Disulfide Bonds in cAlyM

The amino acid sequences of cAlyM and the other characterized PL7 family alginate lyases were aligned to determine the conservative residues. 19 residues were fully conserved, viz., Trp55, Pro60, Thr106, Tyr112, Arg114, Glu116, Leu117, Arg118, Gln183, Ile184, His185, Leu240, Tyr286, Phe287, Lys288, Ala289, Gly290, Tyr292, and Gln294, as shown in Figure 1A. Of these, Gln183, Ile184, and His185 corresponded to the catalytic residues proposed to exist in PL-7 alginate lyases [23,24]. The crystal structure of alginate lyase from *Klebsiella pneumoniae* (PDB code: 4OZX, 100.0% confidence, 98% coverage, 61% identity) was used as the template. Reliability analysis by the Ramachandran plot and Profile-three-dimensional (3D) showed that the model was reliable (Appendix A).

A total of 26 potential disulfide bonds were predicted by the DS software (Appendix A). Based on the analysis of the score value, the disulfide bonds G171C-V176C, S47C-L52C, and N214C-G277C, which showed low scores (<70), were removed. The disulfide bond W55C-L117C was removed, as the amino acid inside the catalytic pocket may hamper enzyme activity. The disulfide bonds I128C-G135C, I184C-P191C, and V248C-V257C were removed, because the distances of the two residues in primary structure were less than 10 amino acids, which may negatively affect the native structure [25]. The disulfide bonds E119C-Y286C, and G182C-L194C, which were located at less than 5 Å with catalytic amino acids, were removed, to ensure the integrity of the catalytic motif [26]. The flexibility of the enzyme was estimated by the root-mean-square deviation (RMSD) values. In comparison, the remaining 17 mutants with overall flexibility were selected as high rigidity mutants and they showed the potential ability to improve enzyme thermal stability [27,28]. Of these, six mutants showed lower RMSD values than those of cAlyM (Table 1). The six mutants were used for further studies, and the positions of the disulfide bonds in the model are shown in Figure 1B.

### 2.2. Determination of Disulfide Bonds of Enzymes

Mutants with disulfide bonds (V59C-Y86C, D102C-A300C, G103C-T113C, R122C-N136C, S173C-S229C, and V207C-I224C) were successfully constructed and expressed in *E. coli* BL21. The molecular weights of the purified cAlyM and its mutants were estimated at approximately 33 kDa by sodium dodecyl sulfate-polyacrylamide gel electrophoresis (SDS-PAGE) (Figure 2).

The content of free sulfhydryl groups in cAlyM under reducing conditions (361.8 µmol/g) kept a similar level to that of under non-reducing conditions (351.1 µmol/g), which means that there was no disulfide bond formed in cAlyM. The contents of free sulfhydryl groups in the mutants V59C-Y86C, D102C-A300C, G103C-T113C, R122C-N136C, and V207C-I224C under reducing conditions were 439.3, 418.4, 421.2, 427.3, and 434.9 µmol/g, respectively, which were higher than those of under non-reducing conditions (385.1, 359.3, 371.6, 379.8, and 382.6 µmol/g). It indicated that new disulfide bonds formed in these mutants. However, mutant S173C-S229C did not form new disulfide bond, according to the content of free sulfhydryl groups, although it introduced more cysteines when compared with cAlyM.

### 2.3. Enzymatic Properties of the Enzymes

When cmpared with cAlyM, the optimal temperature of D102C-A300C, G103C-T113C, and S173C-S229C remained unchanged at 55 °C (Figure 3A). The optimal temperatures of V59C-Y86C, R122C-N136C, and V207C-I224C, were 45 °C, 40 °C, and 50 °C, respectively, and they were lower than the optimal temperature of cAlyM. As shown in Figure 3B, the optimal pH of R122C-N136C remained unchanged as compared with that of cAlyM at 7.0. The optimal pH of V59C-Y86C, D102C-A300C, G103C-T113C, S173C-S229C, and V207C-I224C was 8.0. The enzyme activity of cAlyM and its mutants was determined at the enzyme optimal temperature and pH (Figure 3C). When compared with cAlyM, D102C-A300C, S173C-S229C, and V207C-I224C showed higher enzyme activity. Figure 3D shows the results of enzyme thermal stability. The half-life values at 45 °C (t_1/2, 45°C_) of cAlyM and the mutants V59C-Y86C, D102C-A300C, G103C-T113C, R122C-N136C, S173C-S229C, and V207C-I224C were 1.90 and 0.37, 4.15, 3.06, 0.75, 0.56, and 0.41 h, respectively.

As shown in Table 2, the *K_m_* and *k_cat_* values of cAlyM were 0.37 mg/mL and 762.4 s^−1^, respectively. The *K_m_* and *k_cat_* values of the mutant V207C-I224C were the lowest and highest at 0.21 mg/mL and 921.5 s^−1^, respectively. *k_cat_* and *K_m_* values of D102C-A300C, S173C-S229C, and V207C-I224C were higher than those of cAlyM, indicating a higher catalytic efficiency toward alginate. The Tm values of D102C-A300C and G103C-T113C were 58.9 °C and 57.4 °C, respectively, which were 1.7 °C and 0.4 °C higher than those of cAlyM, whereas the other four mutants showed lower Tm values (Table 2).

### 2.4. Analysis of the Molecular Structure of cAlyM Mutants

RMSD and RMSF values, non-covalent bonds, surface charge, and secondary structure of cAlyM and its mutants were determined to understand the molecular mechanism of increased thermal stability induced by the introduction of disulfide bonds.

The RMSD value of the mutants was higher than that of cAlyM. Moreover, the RMSF value of the mutants was also higher than that of cAlyM, except for G103C-T113C (Table 1). It was observed that the overall structure rigidity of the mutants was higher than that of cAlyM, but the local rigidity of G103C-T113C was lower than that of cAlyM. Table 3 shows the changes in the H-bonds, salt bond, and hydrophobic interactions between cAlyM and its mutants. The number of H-bonds in cAlyM and its mutants (V59C-Y86C, D102C-A300C, G103C-T113C, R122C-N136C, S173C-S229C, and V207C-I224C) were 281, 282, 281, 280, 275, 281, and 281, respectively. Figure 4 shows the changes in the H-bond network within the 5 Å region around the mutant sites between cAlyM and its mutants. The H-bond of S173C-S229C and V207C-I224C remained unchanged. The H-bond of D102C-A300C between D102 and G103 changed to I101 and C300. One native H-bond between P100 and T113 was lost in G103C-T113C. R122C-N136C lost six H-bonds, two between E74 and R122, one between R122 and I128, one between R122 and T130, one between N136 and N137, and one between T130 and N136. One new H-bond was formed between T85 and C86 in V59C-Y86C. When compared with cAlyM, the number of salt bonds in D102C-A300C and R122C-N136C decreased and those in the other mutants remained unchanged. Moreover, when compared with cAlyM, the number of hydrophobic interactions in G103C-T113C and R122C-N136C increased, whereas those in V59C-Y86C and V207C-I224C decreased. As shown in Appendix A, as compared with cAlyM, the surface charge around the mutant sites of G103C-T113C, R122C-N136C, and V207C-I224C remained unchanged. The surface charge around the mutant sites of V59C-Y86C, D102C-A300C, and S173C-S229C was slightly more positive than that of cAlyM.

The secondary structure of cAlyM and D102C-A300C were analyzed while using CD. The CD spectra showed that their secondary structures were almost similar (Figure 5). Both of the enzymes displayed a peak at 195 nm and a valley at 220 nm. The results indicated that both of the enzymes were primarily composed of β-sheets and β-turns. The percentages of α-helix, β-sheet, β-turn, and random coil were 2.53%, 35.76%, 46.89%, and 14.82%, respectively.

## 3. Discussion

Rational designing is a common method of improving the thermal stability of an enzyme. The introduction of disulfide bonds as a type of rational design is an effective strategy for improving protein thermal stability [18]. However, the introduction of inappropriate disulfide bonds may reduce the enzyme activity. An adequate strategy for selecting disulfide bonds is crucial for improving enzyme thermal stability. In this study, the strategy of selected disulfide bonds was based on the analysis of catalytic sites, secondary structure, spatial distance, and 3D-structure flexibility and showed high efficiency. Among the six mutants, there were five mutants that successfully formed new disulfide bonds. 

In previous studies, the alginate lyases have exhibited poor thermal stability, such as alginate lyase PA1167 from *Pseudomonas aeruginosa* (remaining activity 25% after incubation at 45 °C for 10 min) [6], alginate lyase Algb from *Vibrio* sp. W13 (remaining activity 30% after incubation at 40 °C for 30 min) [8], alginate lyase AlyA-OU02 from *Vibrio splendidus* OU02 (remaining activity 35% after incubation at 40 °C for 1 h) [16], alginate lyase PyAly from *Pyropia yezoensis* (t_1/2, 32.5 °C_ of 30 min) [29], and alginate lyase recLbAly28 from *Littorina brevicula* (t_1/2, 47 °C_ of 20 min) [7]. There were also some alginate lyases that exhibited good thermal stability, but their enzyme activities were not high. These included alginate lyase AlgC-PL7 from *Cobetia* sp. NAP1 (remaining activity 90% after incubation at 50 °C for 60 min; optimal enzyme activity 24 U/mg) [30], alginate lyase rSAGL from *Flavobacterium* sp. H63 (remaining activity 98.8% after incubation at 50 °C for 120 min; optimal enzyme activity 4.04 U/mg) [31], and alginate lyase Aly7B_Wf from marine bacterium (remaining activity 38% after incubation at 60 °C for 60 min; optimal enzyme activity 23 U/mg) [32]. The optimal enzyme activities of mutants D102C-A300C and G103C-T113C were 1567.6 U/mg and 754.7 U/mg (determined at OD520), or 1441.4 U/mg and 679.3 U/mg (determined at OD520), respectively. The t_1/2, 45 °C_ of mutants D102C-A300C and G103C-T113C were 4.15 h and 3.06 h, respectively, which were 2.18- and 1.61-times higher than those of cAlyM, respectively. The t_1/2, 50 °C_ and t_1/2, 55 °C_ of D102C-A300C were 2.12 h and 28.9 min, which were 2.02- and 1.9-times higher than those of cAlyM, respectively (data not shown). In addition, the enzyme specific activity of D102C-A300C was 1.13-times higher than that of cAlyM. The catalytic efficiency of D102C-A300C also remarkably improved, with the *k_cat_*/*K_m_* value (3079.2 mL/mg/s) being 149.4% higher than that of cAlyM (2060.7 mL/mg/s), which was higher than that reported by previous studies, such as alginate lyase OalC17 (118.78 mL/mg/s) from *Cellulophaga* sp. SY116 [33] and alginate lyase OalS6 (61.7 mL/mg/s) from *Shewanella* sp. Kz7 [34].

Previous studies have shown that disulfide bonds, H-bonds, salt bonds, and hydrophobic interactions can enhance protein thermal stability [28,35,36,37]. The introduction of disulfide bonds in protein structure is the main reason for improving enzymes’ thermal stability, which can reduce the entropy of the unfolded protein and stabilize the protein conformation. In previous studies, various enzymes’ thermal stability were improved by introducing disulfide bond, such as phytases [20], alkaline α-amylase [21], alginate lyase [26], and 1,4-β-endoglucanase [38]. The newly-formed disulfide bonds D102C-A300C and G103C-T113C that increased the thermal stability of alginate lyase were both located at the protein surface and far from enzyme catalytic center. The formation of new H-bonds increases the thermal stability of cellobiohydrolase Cel7A from *Hypocrea jecorina* [39], alkaline pectate lyase from *Bacillus subtilis* 168 [14], and alginate lyase from the *Flavobacterium* sp. UMI-01 [40]. According to the 3D modeling structure, the H-bonds locus in D102C-A300C changed and hydrophobic interactions in G103C-T113C increased, it might be another reason for the increased thermal stability of the two mutants, in addition to the formation of the disulfide bond. The thermal stability of V59C-Y86C reduced, which is concurrent with a report stating that the disulfide bonds near the N-terminal protein have a negative influence on enzyme thermal stability [26].

cAlyM showed high activity at 45–55 °C, but the activity reduced by more than 50% after incubation at 45 °C for 2 h. The thermal stability of cAlyM cannot meet the industrial demand for the preparation of alginate oligosaccharides. In this study, the alginate lyase D102C-A300C was obtained, which exhibited high thermal stability and enzyme activity by the introduction of a disulfide bond. The optimal temperature and pH of D102C-A300C were 55 °C and 8.0, respectively. The *K_m_* and *k_cat_* values of D102C-A300C were 0.28 mg/mL and 862.2 s^−1^, respectively. The t_1/2, 45°C_ of D102C-A300C was 4.15 h, which was 2.18-times higher than that of cAlyM. The newly formed disulfide and hydrogen bonds may contribute to the increase in enzyme thermal stability. When compared with cAlyM, the introduction of the disulfide bond did not change the secondary structure of D102C-A300C. This rational design method, combined with the prediction of disulfide bonds and screening through the analysis of catalytic sites, secondary structure, spatial configuration, and MD simulation, is expected to be widely applied to enhance the thermal stability of other industrial enzymes. The mutant D102C-A300C showed high potential for the development of functional alginate oligosaccharides in food and medicine industries.

## 4. Materials and Methods

### 4.1. Strains, Media, and Chemicals

The recombinant plasmid harboring the alginate lyase gene, named DH5α-HTa-*cAlyM*, was transformed in *E. coli* DH5α and preserved in our lab. The expression system was *E. coli* BL21 (DE3) cell and pProEX HTa plasmid, which were preserved in our lab. Luria–Bertani (LB) medium comprised of 10 g/L NaCl, 10 g/L tryptone and 5 g/L yeast extract (100 µg/mL ampicillin was added before use). High-fidelity DNA polymerase was purchased from Vazyme Biotech Co., Ltd. (Nanjing, China). The plasmid extraction kit was purchased from Omega Bio-tek, Inc. (Norcross, GA, USA). D*pn*I, a restriction enzyme, was purchased from Thermo Fisher Scientific (Waltham, MA, USA). 

### 4.2. Computational Analysis of Enzymes

Clustal Omega (https://www.ebi.ac.uk/Tools/msa/clustalo/) was used to align the amino acid sequence of cAlyM with other characterized PL7 alginate lyases. Homology modeling of protein structures was performed while using the Phyre2 program (http://www.sbg.bio.ic.ac.uk/phyre2/html/page.cgi?id=index), and reliability analysis of the model was evaluated by the Ramachandran plot and Profile-3D. Disulfide bonds were predicted by the Predict Disulfide Bridges module of Discovery Studio 2018 (DS2018) (Accelrys, Inc., San Diego, CA, USA). Conjugate gradient algorithm and Standard Dynamics Cascade module of DS 2018 were performed to optimize the structure energy and for MD simulation of enzymes, respectively. MD simulation was composed of five stages, viz., two energy minimizations, heating, equilibration, and production stages. In the first energy minimization stage, maximum steps, and root mean square (RMS) gradient parameters were set as 10,000 and 0.2 for the steepest descent. The maximum steps and RMS gradient parameters were set as 10,000 and 0.0001 for conjugate gradient in the second energy minimization stage. For the heating stage, the temperature was increased from 50 to 300 K during 2000 steps with a time step of 0.002 ps. The enzyme model was then equilibrated at 300 K for 20 ps, and data were sampled at 300 K for 200 ps under constant pressure and temperature dynamics. RMSD and RMSF analyses were performed while using the trajectory analysis module of DS2018. The three-dimensional (3D) molecular structures were visualized while using the PyMOL 2.1.1 software (Delano Scientific, San Carlos, CA, USA).

### 4.3. Construction, Expression, and Purification of cAlyM and Its Mutants

The center of the mutation site and 10–15 bases on both sides were selected as the primer sequence. The primers were designed while using DNAMAN 6.0 (Lynnon Biosoft, San Ramon, CA, USA) (Table 4). DH5α-HTa-*cAlyM*, extracted while using the plasmid extraction kit, was used as the template to amplify the mutation plasmids. The conditions of PCR were, as follows: 2 min at 94 °C, followed by 30 cycles of 10 s at 94 °C, 15 s at 60–68 °C, and 3 min at 72 °C, with a final extension step for 5 min at 72 °C. The restriction enzyme DpnI was used to remove the template DNA. The PCR product was purified by the Gel Extraction Kit. Recombinant plasmids were confirmed by DNA sequencing (Ruibiotech, Qingdao, China) and transformed into *E. coli* BL21 (DE3). *E. coli* BL21 (DE3) cells were cultivated in LB medium with 100 µg/mL ampicillin at 37 °C until the optical density at 600 nm reached 0.6–0.8. Subsequently, 0.5 mM isopropyl β-d-thiogalactopyranoside (IPTG) was added into the medium for induction and the medium was incubated at 23 °C for 24 h. The fermentation liquor was centrifuged at 8000 rmp for 20 min; the supernatant is the crude enzyme. 

The enzyme with the His6 tag were purified while using the Ni-NTA agarose column (Cube Biotech, Germany), which was pre-equilibrated in 50 mM phosphate buffer containing 100 mM NaCl, pH 7.0 (buffer A). The enzymes were fully absorbed by the Ni-NTA agarose column. The His-tagged target protein was then eluted with buffer A containing 10–400 mM imidazole. The purified enzymes were detected by 12% sodium dodecyl sulfate-polyacrylamide gel electrophoresis (SDS-PAGE) and then used for further studies after dialysis and ultrafiltration concentration.

### 4.4. Determination of Disulfide Bonds Formation

The formation of disulfide bonds was detected by detecting free amino acids in enzymes. DTNB can quantitatively determine the amount of free sulfhydryl groups in proteins [41]. Two protein samples (1 mL) were added 1 mL Tris-Gly buffer (pH 8.0), one of which was treated with 8 M carbamide and then incubated at 37 °C for two hours. The treated protein samples were added 50 µL 4 mg/mL DTNB and the absorbance at 412 nm was measured to calculate the content of sulfhydryl.

### 4.5. Enzyme Activity Assays

The enzyme activity was determined by the 3,5-dinitrosalicylic acid (DNS) method while using glucose as the standard [42]. The enzymatic reaction was performed in 50 mM sodium phosphate buffer (pH 7.0) containing 0.5% (w/v) alginate for 10 min. at the optimal temperature. One unit of enzyme (U) was defined as the amount of enzyme causing the release of 1 μmol of reducing sugar from alginate per minute. The protein concentration was determined by the Bradford method while using bovine serum albumin as the standard [43].

### 4.6. Properties of cAlyM and Its Mutant 

The kinetic parameters of cAlyM were evaluated by determining the enzyme activity while using alginate at different concentrations (0.05–2.0 mg/mL). The *K_m_* and *V_max_* values were calculated by the Lineweaver–Burk method. The optimum temperature for alginate lyase activity was determined in 50 mM sodium phosphate buffer, pH 7.0, at various temperatures ranging from 40 °C to 60 °C. The optimum pH for alginate lyase activity was determined at optimum temperature in various buffers (50 mM), including citrate buffer (pH 5.0–6.0), phosphate buffer (pH 6.0–8.0), and glycine-NaOH buffer (pH 9.0). The enzymes were incubated at various temperatures (45–55 °C) for different time periods to determine the thermal stability. The melting temperature (Tm) of the enzyme was determined while using MicroCal PEAQ-DSC Automated (Malvern Panalytical Ltd., Malvern, UK) at an enzyme concentration of 1 mg/mL in 20 mM phosphate buffer (pH 7.4). The temperature was increased from 25 °C to 90 °C at a scan rate of 1 °C/min.

### 4.7. Circular Dichroism (CD) Analysis

The CD spectra of cAlyM and its mutants were analyzed while using a MOS-450 circular dichroism spectrometer (Bio-logic, France), as described previously [44,45]. The Dichroweb online software was used to estimate the percentages of secondary structures (α-helix, β-sheet, β-turns, and loops) [46].

## Figures and Tables

**Figure 1 marinedrugs-17-00378-f001:**
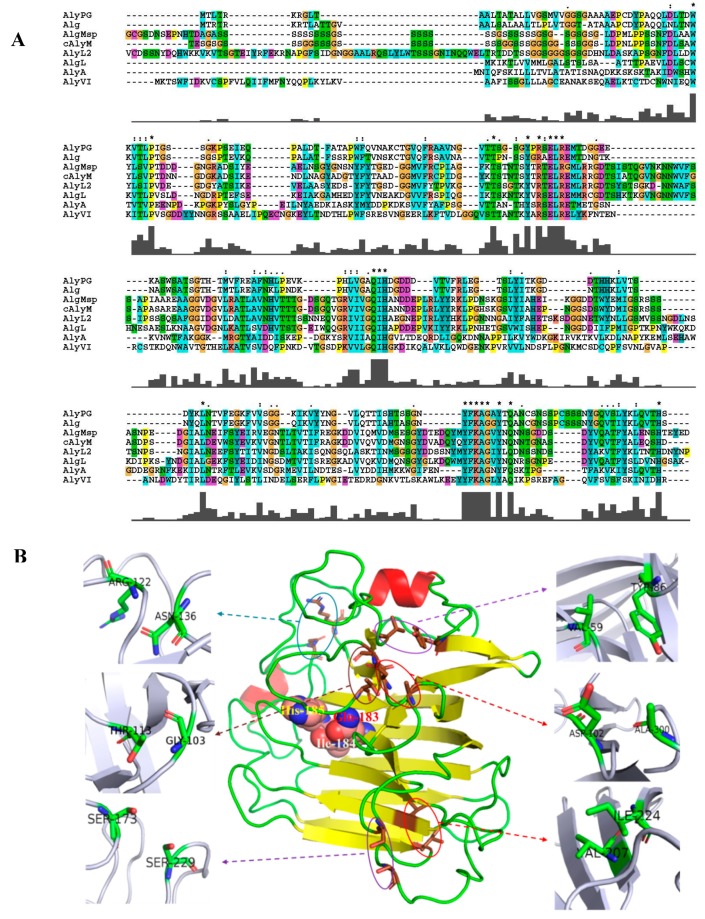
Analysis of the primary and tertiary structure of cAlyM. (**A**) Comparison among the amino acid sequences of cAlyM and several characterized family PL-7 alginate lyases. AlyPG, *Corynebacterium* sp. ALY-1 polyguluronate lyase (GenBank accession no: BAA83339.1); Alg, *Streptomyces* sp. ALG5 alginate lyase precursor (GenBank accession no: ABS59291.1); AlgMap, *Microbulbifer* sp. 6532A alginate lyase (GenBank accession no: BAJ62034.1); AlyL2, *Agarivorans *sp. L11 (GenBank accession no: AJO61885.1); AlgL, *Agarivorans* sp. JAM-A1m alginate lyase (GenBank accession no: BAG70358.1); AlyA, *Flavobacterium* sp. UMI-01 alginate lyase (GenBank accession no: BAP05660.1); AlyVI *Vibrio* sp. QY101 alginate lyase (GenBank accession no: AAP45155.1). Residues invariant among all listed proteins are indicated with an asterisk. Catalytic residues proposed in PL-7 alginate lyases are shown as solid circles. (**B**) The three-dimensional (3D) model of cAlyM and position of the six disulfide bonds (V59C-Y86C, D102C-A300C, G103C-T113C, R122C-N136C, S173C-S229C, and V207C-I224C). Three-dimensional molecular visualization was performed while using the PyMOL 2.1.1 software. In the cAlyM model, the brown amino acids indicate mutation sites, and the arrows point to their magnified structures. The catalytic residues Gln183, Ile184, and His185 are shown according to CPK (Corey–Pauling–Koltun) coloring.

**Figure 2 marinedrugs-17-00378-f002:**
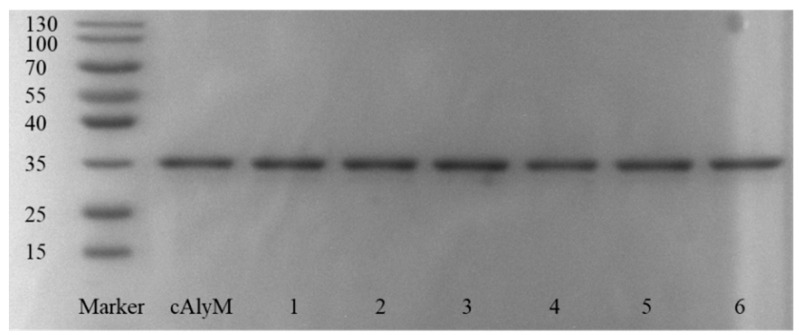
Sodium dodecyl sulfate-polyacrylamide gel electrophoresis (SDS-PAGE) analysis of alginate lyases cAlyM and its mutants. 1, V59C-Y86C; 2, D102C-A300C; 3, G103C-T113C; 4, R122C-N136C; 5, S173C-S229C; 6, V207C-I224C. The enzymes were purified using the Ni-NTA agarose column and detected by SDS-PAGE.

**Figure 3 marinedrugs-17-00378-f003:**
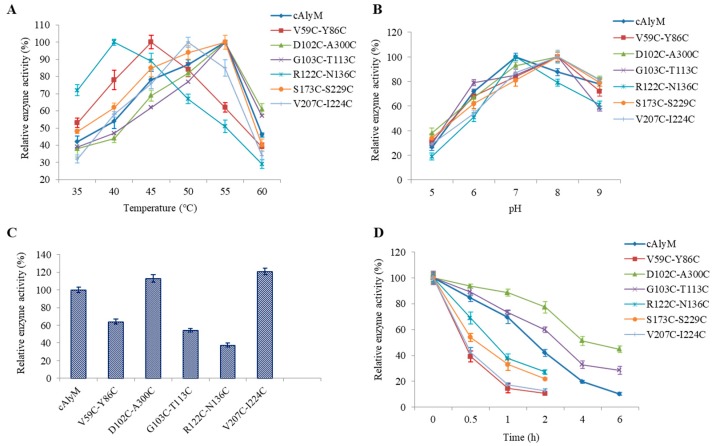
The optimal temperatures (**A**), pH (**B**), enzyme activity at optimal temperature and pH (**C**), and thermal stability at 45 °C (**D**) of cAlyM and its mutants. The thermal stability of cAlyM and its mutants was investigated by measuring the residual alginate lyase activity of the enzyme after incubation at 45 °C for 6 h.

**Figure 4 marinedrugs-17-00378-f004:**
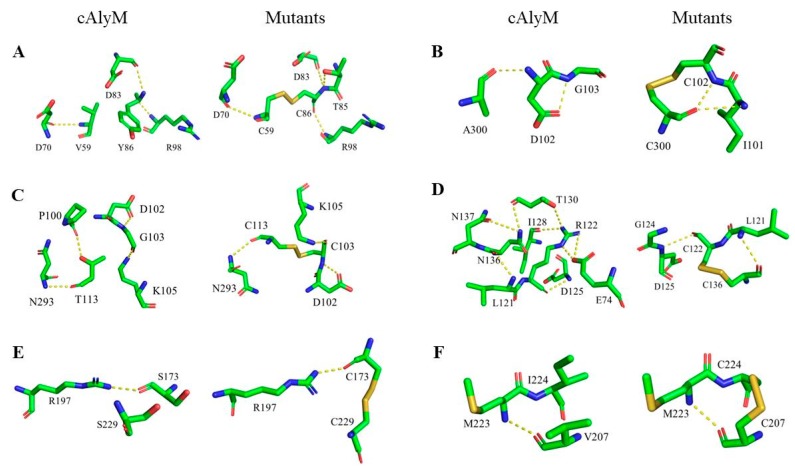
Comparison of the H-bond network within the 5 Å region around the mutant sites between cAlyM and the mutants. (**A**) V59C-Y86C; (**B**) D102C-A300C; (**C**) G103C-T113C; (**D**) R122C-N136C; (**E**) S173C-S229C; (**F**) V207C-I224C.

**Figure 5 marinedrugs-17-00378-f005:**
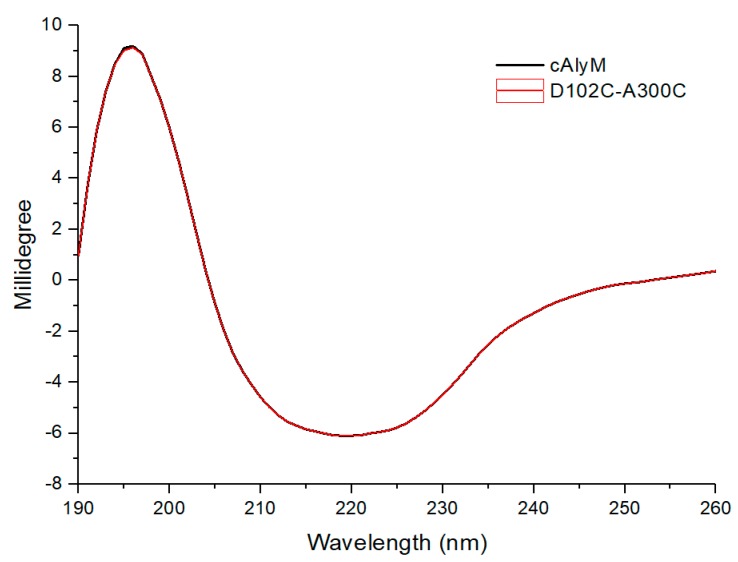
Comparison of the Circular Dichroism (CD) spectra of cAlyM and D102C-A300C. The concentration of enzymes in 20 mM phosphate buffer (pH 7.4) was set at 0.5 mg/mL, and 20 mM phosphate buffer was used as the blank.

**Table 1 marinedrugs-17-00378-t001:** Comparison of the average overall root-mean-square deviation (RMSD) and root mean square fluctuation (RMSF) values of the mutation sites of cAlyM and its mutants.

Mutants	Average Overall RMSD			Mutation Sites RMSF		
	cAlyM	Mutants	RMSD (cAlyM -Mutants)	cAlyM	Mutants	RMSF (cAlyM -Mutants)
V59C-Y86C	1.022	0.925	0.097	0.965	0.921	0.044
D102C-A300C		1.004	0.018	1.277	1.204	0.073
G103C-T113C		0.996	0.026	0.868	0.994	−0.126
R122C-N136C		1.003	0.019	0.921	0.869	0.052
S173C-S229C		0.989	0.033	1.772	1.201	0.571
V207C-I224C		0.962	0.060	0.856	0.823	0.033

**Table 2 marinedrugs-17-00378-t002:** Kinetic parameters and Tm value of cAlyM and its mutants.

Enzymes	*K_m_* (mg/mL)	*V_max_* (U/mg)	*K_cat_* (s^−1^)	*K_cat_*/*K_m_* (mL/s/mg)	Tm (°C)
cAlyM	0.37 ± 0.09	1386.3 ± 23.5	762.4 ± 19.6	2060.7	57.2
V59C-Y86C	0.95 ± 0.21	1166.9 ± 19.2	641.8 ± 18.5	675.6	54.3
D102C-A300C	0.28 ± 0.04	1567.6 ± 52.4	862.2 ± 30.3	3079.2	58.9
G103C-T113C	1.26 ± 0.32	754.7 ± 16.8	415.1 ± 15.5	329.4	57.6
R122C-N136C	1.94 ± 0.28	521.6 ± 17.2	286.9 ± 14.4	147.9	55.6
S173C-S229C	0.31 ± 0.08	1455.1 ± 42.8	800.33 ± 32.3	2581.7	55.4
V207C-I224C	0.21 ± 0.07	1675.4 ± 37.2	921.5 ± 28.6	4387.9	55.4

**Table 3 marinedrugs-17-00378-t003:** The number of bonds in cAlyM and its mutants.

Mutants	cAlyM	V59C-Y86C	D102C-A300C	G103C-T113C	R122C-N136C	S173C-S229C	V207C-I224C
H-bond	281	282	281	280	275	281	281
Salt-bond	26	26	25	26	24	26	26
Hydrophobic interaction	120	117	120	126	121	120	118

**Table 4 marinedrugs-17-00378-t004:** Primers used in this study.

Primer Name	Sequence (5’-3’)
V59C-F	GGTACCTGAGCTGTCCTACCGACAAC
V59C-R	GTTGTCGGTAGGACAGCTCAGGTACC
Y86C-F	CAGATGGCACCTGCTTCTATACTGCTG
Y86C-R	CAGCAGTATAGAAGCAGGTGCCATCTG
D102C-F	GCTGCCCGATCTGTGGCTATAAAAC
D102C-R	GTTTTATAGCCACAGATCGGGCAGC
A300C-F	CACCGGCAATTGCAGTGACTATGTC
A300C-R	GACATAGTCACTGCAATTGCCGGTG
G103C-F	GCCCGATCGATTGCTATAAAACATCG
G103C-R	CGATGTTTTATAGCAATCGATCGGGC
T113C-F	CACGTCCTATTGCCGCACCGAGCTG
T113C-R	CAGCTCGGTGCGGCAATAGGACGTG
R122C-F	CGCGAGATGCTATGTCGTGGCGACACC
R122C-R	GGTGTCGCCACGACATAGCATCTCGCG
N136C-F	GGGTCAATGGATGCAACTGGGTATTCG
N136C-R	CGAATACCCAGTTGCATCCATTGACCC
S173C-F	CTACCGGAGATTGCGGCCAGGTTGGAC
S173C-R	GTCCAACCTGGCCGCAATCTCCGGTAG
S229C-F	GGCAGCCGTTCCTGCAGCGCCTCGGAC
S229C-R	GTCCGAGGCGCTGCAGGAACGGCTGC
V207C-F	GCAAAGGTTCTTGCTATATCGCCCATG
V207C-R	CATGGGCGATATAGCAAGAACCTTTGC
I224C-F	GGTACGACATGTGTGGCAGCCGTTCC
I224C-R	GGAACGGCTGCCACACATGTCGTACC

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
