# Peer review of "Rational Design of Alginate Lyase from Microbulbifer sp. Q7 to Improve Thermal Stability"

_marinedrugs, 2019, doi:10.3390/md17060378_

Round 1
Reviewer 1 Report
This article describes the trial to improve the thermal stability of alginate lyase from Microbulbifer sp. Q7. The authors claimed that the rational design to introduce the disulfide bonds was efficient to enhance the thermal stability. Although some results are of interest, the most important information is not enough.
1. The activity of alginate lyase has been usually determined by measuring the increase in absorption at 230 nm because the enzyme forms a double bond between C4 and C5 carbons. On line 193, the authors mentioned that some previously reported alginate lyases with good thermal stability did not have high enzyme activities. However, such enzyme activities were determined by measuring the absorption at 230 nm. I think that the authors’ enzyme from Microbulbifer sp. Q7 also should be estimated by the same method.
2. The authors demonstrated that the D102C-A300C enzyme was most stable in the mutant enzymes prepared. However, compared with the wild type enzyme, the number of hydrogen bonds, salt bonds, and hydrophobic interaction remained unchanged in the case of the D102C-A300C enzyme. I think the discussion about the data is not enough.
Author Response
Thank you for your comments. We have carefully considered the comments and have revised the manuscript accordingly. Below we address each of the reviewers’ comments point by point:
1. The activity of alginate lyase has been usually determined by measuring the increase in absorption at 230 nm because the enzyme forms a double bond between C4 and C5 carbons. On line 193, the authors mentioned that some previously reported alginate lyases with good thermal stability did not have high enzyme activities. However, such enzyme activities were determined by measuring the absorption at 230 nm. I think that the authors’ enzyme from Microbulbifer sp. Q7 also should be estimated by the same method.
We have added the enzyme activity determined by the method of measuring OD230 nm. The optimal enzyme activity of mutants D102C-A300C and G103C-T113C were 1567.6 U/mg and 754.7 U/mg (determined at OD520), or 1441.4 U/mg and 679.3 U/mg (determined at OD520), respectively (Lines 200-202).
2. The authors demonstrated that the D102C-A300C enzyme was most stable in the mutant enzymes prepared. However, compared with the wild type enzyme, the number of hydrogen bonds, salt bonds, and hydrophobic interaction remained unchanged in the case of the D102C-A300C enzyme. I think the discussion about the data is not enough.
We have revised the discussion in lines 212-218. The introduction of disulfide bonds in protein structure is the main reason to improve enzymes’ thermal stability, which can reduce the entropy of the unfolded protein and stabilize the protein conformation. In previous studies, various enzymes’ thermal stability improved by introducing disulfide bond, such as phytases [19], alkaline α-amylase [20], alginate lyase [25] and 1,4-β-endoglucanase [38]. The newly-formed disulfide bonds D102C-A300C and G103C-T113C that increased the thermal stability of alginate lyase were both located at the protein surface and far from enzyme catalytic center.
Reviewer 2 Report
The ms of Min Yang et al describes the construction of modified alginate lyase by introducing the disulfide bonds into the molecule. The paper is well written, and can be accepted after minor revision.
Some questions.
This is not clear why alginate lyase cAlyM was choosen as a model enzyme, may be it has some advantages in compare with other enzymes?
Fig 3 - please show curves corresponding to cAlyM with higher thickness, this is hard to distinguish it from mutants.
Table 2 - to me the diffrences in the Tm values are not significant, I would say Tm values of all proteins are similar.
please provide the list of plasmids. This is also not clear what vector is used for overexpression of the protein. Have you cloned the gene in this work or in previous one?
The mutagenesis desing is written very poor. As I can understand you made self-complementary primers with modified sequence inside and amplified the whole plasmid, as described for QuikChange II Site-Directed Mutagenesis Kit (Agilent). I recommend either mention the kit used, or describe more detailed protocol.
This is not clear why only phosphate buffer was used for optimal pH assaya, since this buffer works in pH range of 5.8-8.0? At lower or higher pH it does not keep the pH.
Finally, I would recommend to strengthen the conclusion.
Author Response
Thank you for your comments. We have carefully considered the comments and have revised the manuscript accordingly.
1. This is not clear why alginate lyase cAlyM was chosen as a model enzyme, may be it has some advantages in compare with other enzymes?
We have studied the effect of conserved domains reconstruction on enzymes properties [21]. The results showed that the conserved domain reconstruction form cAlyM showed better properties in transcription level, enzyme activity and thermal stability compared other conserved domain reconstruction forms. So the cAlyM was chosen as a model enzyme in this study.
2. Fig 3 - please show curves corresponding to cAlyM with higher thickness, this is hard to distinguish it from mutants.
Thanks for your comment. We have thickened the curves corresponding to cAlyM in Fig 3.
3. Table 2 - to me the differences in the Tm values are not significant, I would say Tm values of all proteins are similar.
Thank you for your comment. The Tm values of all enzymes are indeed similar. Since the main purpose of this study was to obtain an enzyme with higher thermal stability. From the results of the thermal stability of all enzymes at 45°C, the t1/2, 45°C of mutants D102C-A300C and G103C-T113C were 4.15 h and 3.06 h, respectively, which were 2.18- and 1.61-times higher than those of cAlyM, respectively. So we want to carry out a systemic analysis on the properties of the mutants to discuss the mechanism.
4. Please provide the list of plasmids. This is also not clear what vector is used for overexpression of the protein. Have you cloned the gene in this work or in previous one?
The plasmid is pProEX HTa. We have cloned the gene cAlyM in previous work [21], the recombinant plasmid named as DH5α-HTa-cAlyM. The expression system was E. coli BL21 (DE3) cell and pProEX HTa plasmid. The corresponding information also added in lines 243-244.
5. The mutagenesis design is written very poor. As I can understand you made self-complementary primers with modified sequence inside and amplified the whole plasmid, as described for QuikChange II Site-Directed Mutagenesis Kit (Agilent). I recommend either mention the kit used, or describe more detailed protocol.
We have added more detailed protocol in this manuscript (lines 272-277). DH5α-HTa-cAlyM, extracted using the plasmid extraction kit, was used as the template to amplify the mutation plasmids. The conditions of PCR were as follows: 2 min at 94°C, followed by 30 cycles of 10 s at 94°C, 15 s at 60–68°C, and 3 min at 72°C, with a final extension step for 5 min at 72°C. The restriction enzyme DpnI was used to remove the template DNA. The PCR product was purified by Gel Extraction Kit. Recombinant plasmids were confirmed by DNA sequencing (Ruibiotech, Qingdao, China) and transformed into E. coli BL21 (DE3).
6. This is not clear why only phosphate buffer was used for optimal pH assay, since this buffer works in pH range of 5.8-8.0? At lower or higher pH it does not keep the pH.
Actually, we used different buffer to determine the optimum pH. The optimum pH for alginate lyase activity was determined at optimum temperature in various buffers (50 mM) including citrate buffer (pH 5.0–6.0), phosphate buffer (pH 6.0–8.0), and glycine-NaOH buffer (pH 9.0). Sorry to miss this information, we have revised it in manuscript (lines 306-309).
7. I would recommend to strengthen the conclusion.
Thanks for your comment. We have revised the conclusion section.
Round 2
Reviewer 1 Report
The revised MS answered adequately my questions. But the sentence about the stability of D102C-A300C (lines 222-224) is wrong, because H-bonds in D102C-A300C did not changed compared with the wild enzyme, cAlyM as shown in Table 3. So, please correct the sentence.
Author Response
Thank you for your comment. The number of H-bond in D102C-A300C indeed did not changed compared with the wild enzyme. But the locus of H-bond in D102C-A300C changed. We also explained it on lines 154-155.The H-bond of D102C-A300C between D102 and G103 changed to I101 and C300. To avoid ambiguity, we have revised the sentence on lines 221-223. According to the 3D modeling structure, H-bonds locus in D102C-A300C changed and hydrophobic interactions in G103C-T113C increased, it might be another reason for the increased thermal stability of the two mutants in addition to the formation of disulfide bond.